# ASTROVISBENCH: A Code Benchmark for Scientific Computing and Visualization in Astronomy

**Sebastian Joseph**[1], **Syed Murtaza Husain**[1], **Stella S. R. Offner**[1], **Stéphanie Juneau**[2],
**Paul Torrey**[3], **Adam S. Bolton**[4], **Juan P. Farias**[1], **Niall Gaffney**[5], **Greg Durrett**[1], **Junyi Jessy Li**[1]

[1] The University of Texas at Austin
[2] NSF National Optical-Infrared Astronomy Research Laboratory
[3] University of Virginia
[4] SLAC National Accelerator Laboratory
[5] Texas Advanced Computing Center
https://astrovisbench.github.io/

## Abstract

Large Language Models (LLMs) are being explored for applications in scientific research, including their capabilities to synthesize literature, answer research questions, generate research ideas, and even conduct computational experiments. Ultimately, our goal is for these to help scientists derive novel scientific insights. In many areas of science, such insights often arise from processing and visualizing data to understand its patterns. However, evaluating whether an LLM-mediated scientific workflow produces outputs conveying the correct scientific insights is challenging to evaluate and has not been addressed in past work. We introduce AStroVisBench, the first benchmark for both scientific computing and visualization in the astronomy domain. AStroVisBench judges a language model's ability to both (1) create astronomy-specific workflows to process and analyze data and (2) visualize the results of these workflows through complex plots. Our evaluation of visualizations uses a novel LLM-as-a-judge workflow, which is validated against annotation by five professional astronomers. Using AStroVisBench we present an evaluation of state-of-the-art language models, showing a significant gap in their ability to engage in astronomy research as useful assistants. This evaluation provides a strong end-to-end evaluation for AI scientists that offers a path forward for the development of visualization-based workflows, which are central to a broad range of domains from physics to biology. We release the code and data for AStroVisBench at `astrovisbench.github.io`.

## 1 Introduction

As large language models evolve, they hold increasing promise as assistants in scientific research to synthesize literature [29, 11] and generate or execute research ideas [32, 25, 14]. However, end-to-end AI science is still seriously lacking [23], as indicated by benchmarks that evaluate models' capabilities to deploy research in computer science [38] and machine learning [33], as well as coding for scientific problems [34].

Useful AI assistants that can aid expert-led scientific progress need to have deep domain knowledge to *implement scientific workflows*: understanding the scientists' queries in the context of their workflow, knowing when and how to use domain-specific APIs, navigating data sources, manipulating and visualizing data for analyses. Although there are challenging code benchmarks, e.g., SWE-bench [16] and BigCodeBench [45], evaluating whether an LLM-mediated scientific workflow produces

39th Conference on Neural Information Processing Systems (NeurIPS 2025) Track on Datasets and Benchmarks.

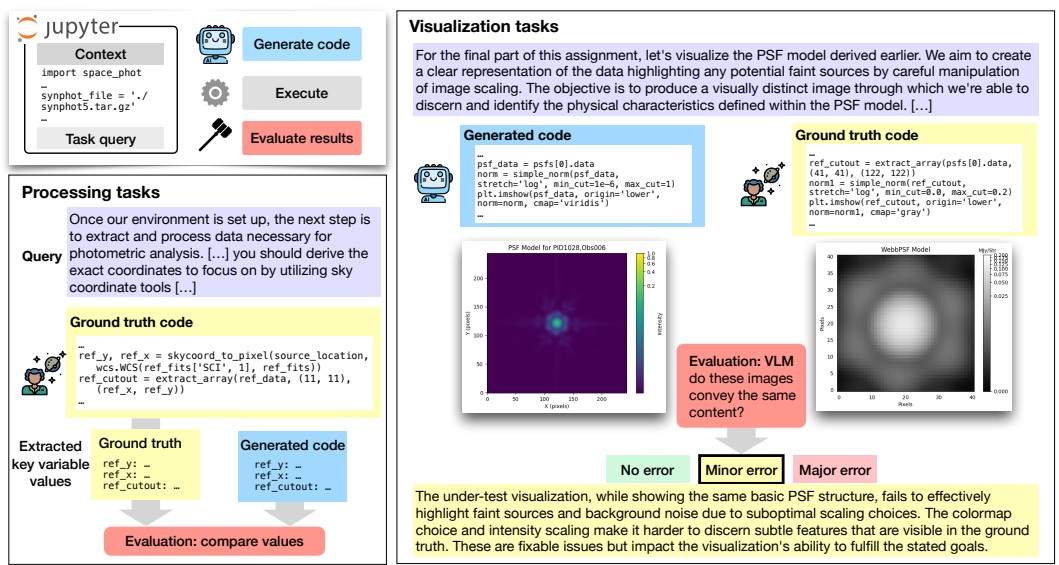

Figure 1: An overview of AsTROVISBENCH, evaluating astronomy research workflow implementation that leads to a visualization. In a Jupyter notebook environment, given a task query $Q$ and the task context (i.e., code prior to the cell-under-test), a subject LLM generates code $\hat{c}$ that is validated as to whether it correctly leads to the right visualization (Section 2.1). There are two types of tasks: **processing tasks** $t_{process}$ involve scientific computing necessary to prepare for the visualization, and **visualization tasks** $t_{visualize}$ involve code that creates a visualization (Section 2.2). Processing tasks are evaluated by executing the ground truth and generated code, and comparing the values of key products necessary for the visualization (Section 3.1). Visualization tasks use a VLM-as-judge we show to correlate highly with expert judgments from professional astronomers (Section 3.2).

visualizations that convey the correct scientific insights is challenging to evaluate and has not been addressed in past work.

This work presents AsTROVISBENCH, a benchmark targeting LLM's capabilities to implement research workflows that result in complex scientific visualizations in the astronomy domain. We choose astronomy as the target domain because of three important characteristics. First, it is a data intensive discipline with heavy reliance on, e.g., database queries, data manipulation, advanced physics and mathematics, data visualization, and simulation modeling. Second, the astronomy research ecosystem is based on large publicly available datasets and open source code, which creates a rich landscape of research-level LLM training and testing material. Third, in contrast to some other academic disciplines, the astronomy community is modest in size and relatively coherent in research focus, meaning that any developed scientific LLM package or benchmark will apply to a large fraction of the community.

Importantly, **scientific visualization** is a critical part of astronomy research. As an observational science, astronomy has a rich history of data exploration via imaging, which dates back to the hand-drawn sketches of the Moon, stellar clusters, and Jupiter recorded by Galileo in the first telescope observations [9]. Present-day astronomical data are high-dimensional and often use heat maps to illustrate spatially and spectrally varying properties of light [10]. These charts vary substantially from those in standard chart understanding datasets from the machine learning literature [27, 37, 4, 43]. More critically, the evaluation of LLMs for scientific visualizations has not assessed their performance at the end-to-end process of producing visualizations.

AsTROVISBENCH contains 864 tasks sourced from 110 Jupyter notebooks that were curated by the NSF National Optical-Infrared Astronomy Research Laboratory (NOIRLab) and the Space Telescope Science Institute to work with data taken by ground-based and space-based observatories, respectively. Intended as tutorials and example use cases, the notebooks span a range of astronomical research applications from simple to advanced tasks and feature different types of data (catalogs, spectra, images). These notebooks illustrate research workflows from data processing to visualization.

Given a query within a Jupyter notebook, the task of an LLM is to generate code to implement the query using the context of the previous cells in the notebook. Our queries are synthesized from LLMs to reflect the style of query that an astronomer would pose if they did not know the precise step-by-step details in accomplishing a task, unlike other potential sources of natural language annotation such as code comments. The benchmark consists of two types of tasks, each with a novel evaluation pipeline that we introduce:

(1) **Processing tasks**: to evaluate code for data analyses leading up to the visualization, we use execution-based evaluation that directly compares the values of key variables that visualizations would depend on, after executing the LLM-generated code with the ground truth values.

(2) **Visualization tasks**: to evaluate the LLM-generated code for visualizations, we engaged six domain experts (post-PhD researchers and faculty members) to establish a set of 270 gold-standard judgments of visualization quality. We further develop an LLM-as-judge that achieves a high correlation with the expert judgments, serving as the automatic evaluator.

When executed on AstroVisBench, even the most advanced models have difficulty in accurately completing the tasks posed in this benchmark, revealing a significant gap in their ability to helpfully engage in these domain-specific visualization-based workflows. In particular, we found many of these models lack the knowledge necessary to use niche, domain-specific libraries and APIs, and they also lack the ability to visualize results in a manner consistent with research standards in the astronomy domain.

## 2   Benchmark Construction

In constructing AstroVisBench, our main consideration for evaluation is testing whether a subject model (e.g., an LLM) can act as a useful coding assistant to correctly perform scientific tasks in astronomy, from data processing to visualization. We address the following challenges: (1) developing a methodology for evaluating the success of accomplishing research tasks with specific fine-grained aims, especially those involving visualization, and (2) ensuring the tasks are representative of *typical-use* interaction between an expert astronomer and an LLM.

### 2.1   Task Setup

We collect code from Jupyter notebooks, which contain text in markdown cells, code in executable cells, and visualizations as figures rendered in the same document. We denote by $C = c_1, ..., c_k$ the set of notebook code cells in a notebook. We select notebooks such that cell $c_k$ always contains code that produces a visualization when executed, and the previous cells $c_1, ..., c_{k-1}$ are extracted dependencies of $c_k$ (i.e., they are not necessarily consecutive in the original notebook, see Figure 2).

We define a visualization pipeline as a set of tasks $\mathbf{T} = (t_{\text{process}}, t_{\text{visualize}})$ given setup code (such as import statements and environmental variables). Each $t_{\text{task}}$ is a bundle $(c_{1...j}, c_{j...k}, q, y)$, where $c_{1...j}$ denotes setup cells prior to the current task, $c_{j+1...k}$ denotes the "core" cells for the task, $q$ is a natural language query expressing the functionality of the core cells, and $y$ is an expected result. Our task is for an LLM to compute $\hat{c} = M(c_{1...j}, q)$, a code cell predicted from the LLM. We then evaluate $v(\texttt{Exec}(c_{1,...,j}, \hat{c}), y)$ with a validator that compares the results of executing our predicted code $\texttt{Exec}(c_{1,...,j}, \hat{c})$ with an expected result $y$.

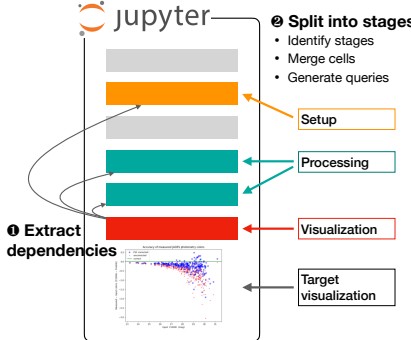

Figure 2: To construct benchmark tasks, we trace dependencies from visualization cells within a notebook, and split these dependencies into three stages, jointly merging these original cells and generating queries for each stage.

## 2.2 Sourcing and Construction

**Notebook Collection**    We gathered notebooks from collections curated by the Astro Data Lab[1] [8, 28] and by the Space Telescope Science Institute.[2] Both publicly available notebook collections are developed to serve as tutorials and example scientific use cases for a target audience composed of astronomy graduate students and professional researchers [17]. While spanning a broad range in difficulty levels, the notebooks are detailed and feature tasks such as querying databases (with SQL and astronomy-specific ADQL), reading astronomical data (tables, spectra or images), performing data processing and analysis, and creating visualizations suitable for publication in scientific journals. In Section 4, we discuss and showcase the diversity of the visualization types present in this collection.

**Visualization Task Extraction**    The code in these Jupyter notebook cells is a mixture of code for data analysis and code for visualizing the results of these analyses. This structure, where visualization cells depend on data analysis, suggests that a Jupyter notebook can be split into several stages:

- **Setup**: This stage involves importing the necessary libraries and other environment setup necessary to proceed with the task (not evaluated).
- **Processing** $t_{\text{process}}$: This is where the core of the data analysis prior to visualization occurs.
- **Visualization** $t_{\text{visualize}}$: This is where the visualization is generated.

We adopt this setup to avoid overly fine-grained or trivial cells present in the notebooks. We use `gpt-4o` to simultaneously split notebooks into these clearly defined stages, and produce natural language queries for each stage (prompt in Appendix B).

**Query Generation Desiderata**    ASTROVISBENCH evaluates the ability of a model to respond to *typical-use* queries from astronomers as an AI assistant (without step-by-step specifics), as if it were an expert itself. This is distinct from the explanations sometimes present in the cells, which are typically explanations rather than queries.

Importantly, we need to ensure that queries do not leak information and domain knowledge to the model that would be expected for an expert in astronomy to be aware of. Our team of expert astronomers verified a subset of the generated queries while inspecting the generated visualizations (Section 3.2.1), in which they confirm that these queries do not leak such information and represent typical research queries.

The natural language queries also need to be specific enough to enable consistent and reliable evaluation. However, critical *underspecifications* within queries, such as the omission of the name of a data file or the omission of a subjective threshold used to filter data, can make this difficult. Therefore, we use `gpt-4o` again to map underspecified spans within a query to numerical values and string literals in the ground truth code. In our evaluation of LLMs using ASTROVISBENCH (See Section 5), we append these underspecification clarifications to the processing query when generating responses from our target LLMs (see Figure 1, "Processing tasks").

## 3   Evaluating Generated Code

With the natural language queries eliciting our subject model to respond with code, the next step in the benchmarking process lies in the evaluation of that code. While accuracy is critical in scientific computing and visualization, it is important to stress that there can be multiple ways to perform an analysis, and a figure can be correct even when it differs from the ground truth, e.g., it may adopt different symbols, colors, or axis ranges. Therefore, we opt for execution-based evaluation rather than those based on surface-level code similarity [31, 44].

Our evaluation setup is depicted in Figure 1. First, we check if *the code executes without error*. If it does the execution results will be passed on to different validators depending on the type of task.

### 3.1   Evaluating $t_{\text{process}}$: Variable Inspection

The processing stage is where most of the scientific computing and data manipulations take place, in preparation for data visualization. In ASTROVISBENCH, such processing includes operations

---

[1] `https://github.com/astro-datalab/notebooks-latest`
[2] `https://spacetelescope.github.io/notebook-infrastructure/`

on numerical data from catalogs (e.g., filtering, regression, supervised classification with a labeled training set, unsupervised classification with clustering) as well as operations on astronomical images (e.g., astrometric alignment, convolution or smoothing, coaddition, mosaic creation, extraction of photometric fluxes, shape determination, morphological classification). Additional processing may include cosmological calculations to infer physical sizes and distances or more advanced computations such as the two-point correlation function or Fourier decomposition.

Since the resulting program state of the processing stage entails variables that the visualization stage needs, we view the intersection of the variables created/modified during $t_{\text{process}}$, and the variables that the corresponding $t_{\text{visualize}}$ is dependent on, as the *key products* of the processing stage. Specifically, for each processing task, we define the key products in the ground truth code as $\mathcal{V}_G$, and the set of variables that received an assignment in the generated code as $\mathbf{V}_M$. We then report the recall of $\mathbf{V}_M$ as the variable inspection score: VIscore $= |\mathbf{V}_M \cap \mathcal{V}_G|/|\mathcal{V}_G|$.

## 3.2 Evaluating $t_{\text{visualize}}$: Expert-informed Visualization Evaluation

To evaluate the code that produces the visualization, we execute the code and visually compare the generated and ground truth visualizations to determine whether the generated visualization conveys the same key information using domain-specific standards of quality as the ground truth visualization.

The ideal judges for this task are astronomy researchers. However, human expert evaluation is not scalable and is therefore impractical for a benchmark. Instead, we automate this process by deploying a VLM as a judge to compare the generated visualization to the ground truth (Section 3.2.2). To guide and validate the VLM judge, we first collected gold-standard quality judgments from professional astronomers (Section 3.2.1).

**Evaluation Setup**   The quality of a generated visualization is judged given: (1) the visualization query, (2) the ground truth visualization and its code, (3) the LLM-generated visualization and its code. Each judge evaluates the severity of errors using the following categories:

- **No Error (1)**: The LLM-generated visualization conveys the same key information as the ground truth visualization.

- **Minor Error (2)**: The LLM-generated visualization could be fixed by making minor adjustments to the code or by clarifying underspecified details in the visualization query.

- **Major Error (3)**: The LLM-generated visualization deviates severely from the ground truth visualization, ultimately conveying very different information from what is being asked in the visualization query.

In addition to labeling the LLM-generated visualization according to the above classes, each judge provides a short explanation of the determination.

### 3.2.1   Expert Evaluation of Visualizations

We worked with five professional astronomers, all of whom have a doctoral degree in astronomy, astrophysics, or physics, and are working as researchers or faculty members. Together with these researchers, we developed the aforementioned evaluation schema and guidelines. In total, we had 6 hours of group discussions, not accounting for individual annotation done asynchronously.

| Type | $\kappa$ ($\uparrow$) | Pairwise $\rho$ |
|---|---|---|
| Error Category | 0.53 | 0.69 |
| Preference | 0.44 | – |

Table 1: Collective and pairwise human expert agreement as measured through Fleiss' $\kappa$ and Spearman's $\rho$ on 30 visualization tasks. Correlation is significant at $p < 1e^{-29}$.

For each ground truth visualization, we provide two LLM-generated visualizations (produced by two different LLMs listed in Section 5) for our experts to evaluate.[3] In addition to determining error categories for each visualization individually, the experts also provided a preference judgment for the pair of generated visualizations for each query, if both fall under the same error category.

In total, our experts evaluated $135\times2$ LLM-generated visualizations. Out of these 135 tasks, 30 were annotated by all five experts to calculate agreement (Table 1). Overall, we see moderate agreement

---

[3]The experts also had access to the original Jupyter notebooks from which the query task was constructed.

| Field | Library |
|---|---|
| Spectroscopy | `specutils, astropy.modeling, astropy.io.fits, scipy.optimize, scipy.stats, lmfit, dust_extinction, astroML` |
| Photometry | `photutils, lightkurve, astroquery.mast, astropy.table, astropy.stats, space_phot, regions` |
| Image Processing | `numpy, scipy.ndimage, astropy.nddata,astropy.convolution, PIL, cv2, astrocut, wfc3tools, jwst` |
| Time Series Analysis | `astropy.timeseries, lightkurve, pandas, scipy.signal, gatspy` |
| Cosmology & Large-Scale Structure | `astropy.cosmology, healpy, astropy.coordinates, reproject, shapely` |
| Simulation & Modeling | `astropy.modeling, keras, tensorflow, sklearn, webbpsf, acstools, stistools, refstis` |

Table 4: API distribution across astronomy fields within ASTROVISBENCH.

on error category assignments, but lower agreement on preference judgments. This is expected, as these preferences are subjective depending on the annotator's background and aesthetic preferences.

### 3.2.2 Automatic Evaluation of Visualizations

We use a VLM as the automatic evaluator for visualizations created from LLM-generated code. The prompt of the VLM is shown in Appendix B.1. Table 2 shows the Spearman correlation between each VLM we evaluated and expert judgments (for the 30 tasks in the agreement set) `Claude 3.5 Sonnet` achieved the highest correlation, hence it used as the automatic evaluator for visualization tasks in ASTROVISBENCH.

## 4 Benchmark Statistics

The benchmark consists of 432 processing and 432 visualization tasks, extracted from 110 Jupyter notebooks. On average, each task covers 6.2 cells in the original notebooks. We present the average token count for every task query and for the ground truth code for each stage in Table 3.

**API Diversity** Table 4 shows the type of libraries/APIs represented in ASTROVISBENCH and their relation to subfields and tasks within astronomy and astrophysics. We cover 38 libraries specialized for scientific and visualization use-cases, with 26 of these libraries designed for astronomy in particular. We provide a more detailed breakdown along with function calls in Appendix A.

| Model | $\rho$ (avg) | $\rho$ (maj) |
|---|---|---|
| Gemini 2.5 Flash | 0.816 | 0.769 |
| Gemini 2.0 Flash | 0.753 | 0.775 |
| Claude 3.7 Sonnet | 0.723 | 0.714 |
| Claude 3.5 Sonnet | **0.822** | **0.828** |
| Claude 3.5 Haiku | 0.749 | 0.586 |

Table 2: The Spearman correlations between vLLM judges and expert judges (averaged scores or majority labels). Correlations are significant ($p < 1e^{-29}$).

| Type | Field | Avg # Token |
|---|---|---|
| Query | Setup | 102.04 |
| | Processing | 108.69 |
| | Visualization | 107.05 |
| Ground Truth | Setup | 87.80 |
| | Processing | 373.96 |
| | Visualization | 116.44 |

Table 3: Number of tokens in the queries and ground truth code in ASTROVISBENCH.

**Processing Products** In the variable inspection test we perform for evaluating the processing tasks, the key products created in this section are stored in a pickled format. This approach can store most Python objects. However, generator objects, lambda functions, nested functions, and objects that hold OS-level resources, such as file descriptors and network sockets, cannot be pickled. We inspected the pickled key products in 359 processing tasks (spanning 101 Jupyter notebooks). These 886 key products span 47 different data types. For each processing task, the average number of key products is 2.46.

**Types of Visualizations** ASTROVISBENCH covers a diverse collection of domain-specific visualizations with a sample of these visualizations shown in Figure 3. The visualizations featured astronomy specific data with several varied data types and plotting modalities. Data types included catalogs of object information (e.g., paired sets of galaxy visual color and brightness), one-dimensional datasets (e.g., time series information about stellar luminosity as a function of time, or spectral energy distributions indicating object intensity as a function of wavelength), and image data with

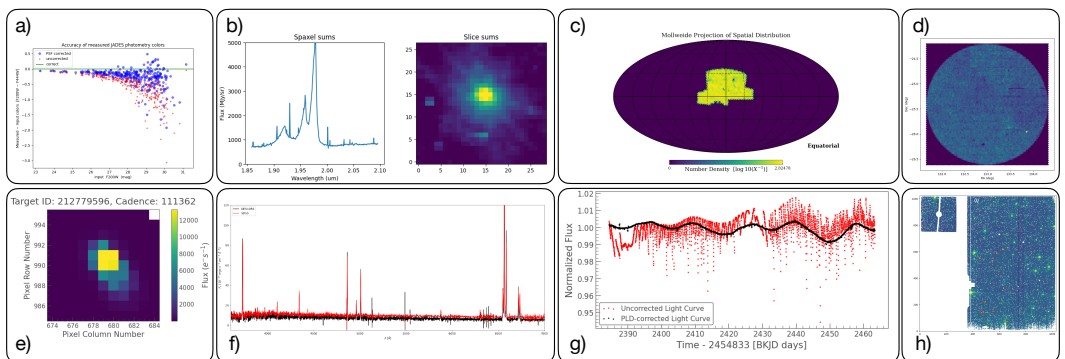

Figure 3: Examples of visualizations in ASTROVISBENCH showing (a) a color-magnitude diagram before and after point spread function correction, (b) a spatially integrated spectral energy distribution (left) and associated spatially resolved intensity map (right), (c) a wide-field all-sky projection of galaxy source counts within a survey footprint, (d) a Kepler mission source pixel map, (e) a pixel-level flux map, (f) galaxy spectra featuring bright emission lines, (g) corrected and uncorrected time-series light curves, and (h) a wide-field image.

| Model | Processing | | Visualization | | | | |
| --- | --- | --- | --- | --- | --- | --- | --- |
| | Crash % | VIscore | Crash % | VisFail % | NoE % | MiE % | MaE % |
| Gemini 2.5 Pro | **30.8** | 0.600 | **20.1** | 9.3 | **15.7** | 25.9 | 28.5 |
| Claude Sonnet 3.7 | 50.9 | 0.633 | 34.7 | 21.3 | 9.5 | 14.6 | 19.2 |
| Claude Opus 4 | 50.3 | 0.644 | 33.1 | 12.7 | 9.3 | 28.9 | 16.0 |
| o3-mini | 51.4 | **0.694** | 30.3 | **2.8** | 10.4 | 26.4 | 29.6 |
| GPT-4o | 53.7 | 0.480 | 32.6 | 6.0 | 8.6 | 23.4 | 28.5 |
| QwQ | 64.1 | 0.472 | 60.9 | 3.9 | 8.2 | 12.6 | 14.9 |
| Qwen-2.5 | 56.9 | 0.527 | 35.4 | 3.7 | 10.4 | 20.1 | 29.9 |
| Llama-4 Maverick | 55.3 | 0.546 | 28.7 | 9.0 | 9.7 | 21.5 | 30.6 |

Table 5: Results of the LLM evaluation on ASTROVISBENCH for both processing and visualization tasks. We show the percent of test instances that crashed (Crash %) for each type of task, the variable inspection score (VIscore), the percent of instances where the model failed to produce only one visualization (VisFail %), and the breakdown of no error (NoE %), minor error (MiE %) and major errors (MaE %) resulting from the automatic visualization evaluation.

varied levels of calibration. The visualizations themselves required appropriate representation of the underlying dataset with prompt specified constraints. For example, catalog data was in some cases best represented with a scatter plot and in other cases with a two-dimensional histogram. Images were always presented as images, but with varied levels of raw-data pre-processing (e.g., background subtraction) and with requirements that the visualization match the objective (e.g., applying an appropriate image stretch to prominently feature faint objects).

## 5 LLM Performance on ASTROVISBENCH

Using ASTROVISBENCH, we conducted an evaluation of eight state-of-the-art LLMs, covering open-source and proprietary models, with and without advanced reasoning capabilities: `Gemini 2.5 Pro`, `Claude 3.7 Sonnet`, `Claude 4.0 Opus`, `GPT-o3-mini`, `GPT-4o`, `Qwen 2.5 72B`, `QwQ 32B`, and `Llama-4 Maverick` (17Bx128E). We provide detailed descriptions of these models in Appendix D.

### 5.1 Results

**Processing Tasks**   We present the results of $t_{\text{process}}$ on the left side of Table 5. The % of crashed code shows a significant gap in the SOTA LLMs' ability to produce executable code for these tasks. Only `Gemini 2.5 Pro` has managed to produce such code for over half of the processing tasks.

We also see a gap in these models' ability to produce the same processing products as the ground truth when the code it generates does indeed execute without raising errors. This is evident in the results of the variable inspection test, with the highest VIscore among all models being 0.694 from `o3-mini`. Although this indicates that on average the generated code from `o3-mini` produces nearly 70% of the needed processing products, the remaining wrong products will still lead to failures to produce the target visualizations correctly.

Finally, while `Gemini 2.5 Pro` dominated in its ability to produce code that executes without errors, its VIScore was not among the highest. This means that a better ability to generate code that runs does not necessarily always mean a better ability to perform the *right* scientific computing.

**Visualization Results**    The right portion of Table 5 shows the visualization evaluation results. For code execution success, `Gemini 2.5 Pro` again performs the best. The absolute percentages of code that executes are higher than that in processing tasks. This is expected as the models are exposed to more context in the visualization stage, and these sub-tasks usually involve the usage of well-known visualization libraries (e.g., `matplotlib`), unlike the niche, domain-specific libraries prominently featured in most processing sub-tasks. However, the percentage of crashed code is still high.

The columns NoE, MiE, and MaE illustrate that the percentage of cases where the VLM judged the model-produced visualization as correct is much lower than those with errors, and the % of major errors are high. `Gemini 2.5 Pro` leads the other models in terms of producing visualizations judged as no error; however, despite this, and despite it producing code that executes, the percentage of major errors remains high. This also means that at least 58% of the time, the best models today produce code that either does not result in a visualization, or results in visualizations that contain major errors.

## 6   Error Analysis

**Execution Errors**    Around 43% of all the code generated by all the subject LLMs fails to execute without errors. In Figure 4, we show the top ten most frequent execution errors that resulted from running LLM-generated code for both processing and visualization tasks. The largest type of error for processing tasks is the `FileNotFoundError`, which is caused by an attempt to act on a file that does not exist. These may be caused by the LLM hallucinating a file path, sometimes because the model is unable to infer it from the context provided. A less common source of such error is introduced by unaddressed underspecification, despite our effort to clarify them (Section 2.2). However, We estimate that this issue affects only 6% of all the execution errors raised during the evaluation.

Other execution errors for both the processing and visualization tasks arise from a lack of knowledge about how to interact with domain-specific scientific tools. LLMs may hallucinate arguments and function calls, provide the wrong type of input, and incorrectly access the data structures produced by such tools. One common error in the LLM response to processing tasks is the failure to generate appropriate ADQL queries, which are needed to collect data from astronomy

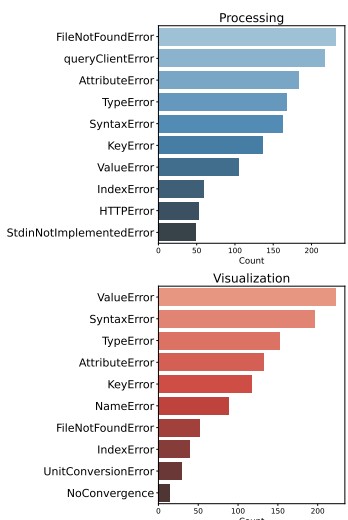

Figure 4: Breakdown on the types and counts of execution errors resulting from the LLM evaluation on both processing and visualization tasks.

databases; these manifest as a `queryClientError`. In visualization tasks, more errors result from interactions with the objects generated during the processing stage, e.g., `ValueError`, `TypeError`, and `KeyError`, as the model fails to understand the internal structure of these objects, often due to function calls from niche astronomy-specific libraries.

**Visualization Errors**    In analyzing the rationales provided by our experts (Section 3.2.1) and in further discussions with them, we identify several common issues in the generated visualizations (with examples shown in Appendix E). First, the experts noticed that the generated visualizations frequently overlook domain-specific plotting conventions, such as the order of the sky coordinates or magnitudes on the axes. Incorrect domain-specific conventions are also evident in an "inappropriate"

choice of axis scale. Another issue flagged by the experts is the failure of the LLMs to judge the qualities of the data it is visualizing. Frequently, the plot ranges unnecessarily cut off part of the data; in visualizations depicting images, the LLMs are not able to optimize to highlight faint or subtle features while de-emphasizing noise. Sometimes, this issue is linked with the LLMs inability to iteratively analyze and tweak its own visualizations like humans, who are able to determine optimal values, such as the stretching scale and range, for their visualizations through such an iterative approach. Finally, experts occasionally flagged issues in the readability of some visualizations, with axis labels and legends having font sizes that were not easily readable.

# 7   Related Work

Recent work has explored automating various parts of scientific research [25, 14], particularly machine learning research and development [13, 41, 22]. However, end-to-end evaluation of research products such as an autogenerated scientific paper that passed workshop peer review has found such work to be seriously lacking [23]. Our work starts from the premise that scientific insights or the ability to assist in particular scientific workflows is a more realistic target for near-term, quantifiable progress. Other work subscribes to a similar view: for instance, in physics, Gravity-Bench-v1 [19] tests an LLM agent's ability for scientific discovery, evaluating how it answers complex, open-ended questions by engaging with a gravity simulation using a few benchmark-specific tools. It differs from ASTROVISBENCH in that our work covers a much more diverse set of scientific workflows in astronomy, requires the use of a large set of domain-specific tools and APIs, and is visualization-focused.

Benchmarks targeting coding tasks currently evaluate models in their general-domain coding capabilities, including solving self-contained tasks [3, 1, 24], using libraries and tools [20, 36, 15], resolving software issues [15], and making diverse function calls [45, 39]. However, scientific research tasks involve much deeper domain knowledge and the ability to perform scientific computing using domain-specific APIs. More recent work like PaperBench and MLEBench [33, 2] targets such capabilities in the machine learning domain. SciCode [34] tests LLMs on their ability to solve computation problems in natural science. However, these are not the workflow tasks targeted in this work, nor do they test the ability to use niche domain-specific research libraries.

Researchers have conducted human evaluations to assess LLM-generated code for charts [6, 35] and visualization design [5, 18]. Automatic evaluation methods have also been developed that span rule-based evaluations [21, 26], self-evaluation [7, 12], and structured LLM-as-a-judge approaches [30, 4] like the one used in ASTROVISBENCH. EvaLLM [30] is one such method that uses an LLM to judge generated Vega-lite visualizations against the ground truth by comparing their JSON representations, yet this approach relies on surface-level code similarity rather than code execution. VisEval [4] uses an execution-based, automatic evaluation method that directly evaluates a visualization as produced from executing LLM-generated code, using a combination of rule-based methods and a VLM. However, they only assess readability, while we focus on scientific utility informed by professional astronomers. In addition to these benchmarks, there have also been work on dedicated solutions for visualization code generation. Zhao et al. [42] tackles a chart-to-code task by synthetically generating a large dataset and using it to fine-tune an LLM. MatPlotAgent [40] is an agentic framework for visualization code generation given a user query and tabular data. ASTROVISBENCH stands out by evaluating how well models support end-to-end research workflows in this domain, which existing methods with standardized data access and no domain focus do not address. Finally, prior work has also evaluated the capacity for models to understand charts [43, 37]. Here, we tackle the opposite problem: chart generation.

We present a table summarizing this related work for easy viewing in Appendix C.

# 8   Conclusion

We present ASTROVISBENCH: the first benchmark for scientific computing and visualization in the astronomy domain. Our work includes the construction of a rich, diverse set of benchmark tasks, the development of an automated framework that directly evaluates the products of execution, including visualizations, and an evaluation of eight state-of-the-art language models, revealing a significant gap in these models ability to engage in astronomy research as useful assistants. This benchmark

paves the way for the future development of models that can aid researchers across a wide range of domains in visualization-based workflows.

**Limitations.** Due to factors of time and cost, we adopt an automatic LLM-based method to aid in constructing ASTROVISBENCH as opposed to manually building it entirely through human experts. This method may introduce noise into the benchmark through hallucinations. However, as experts verify a sizable subset of the benchmark tasks, we are confident that the presence of hallucinations is minimal. Similarly, the automatic evaluation of visualizations in this benchmark heavily relies on a vision LLM, which could make judgments that are not aligned with those of experts. Nevertheless, since a subset of these judgments are well-correlated with those of experts, such judgments are unlikely to significantly influence the overall evaluation. The evaluation of processing products is limited by the inability to store certain runtime objects in a persistent state. Nonetheless, this best-effort method covers a large amount of key objects that informs the ability of LLMs to complete processing tasks correctly.

## Acknowledgments

This work was supported by the National Science Foundation under Cooperative Agreement 2421782 and the Simons Foundation grant MPS-AI-00010515 awarded to the NSF-Simons AI Institute for Cosmic Origins — CosmicAI, `https://www.cosmicai.org/`. This research uses services or data provided by the Astro Data Lab, which is part of the Community Science and Data Center (CSDC) Program of NSF NOIRLab. NOIRLab is operated by the Association of Universities for Research in Astronomy (AURA), Inc. under a cooperative agreement with the U.S. National Science Foundation. The authors acknowledge the Texas Advanced Computing Center (TACC) at The University of Texas at Austin for providing computational resources that have contributed to the research results reported within this paper. URL: `http://www.tacc.utexas.edu`.

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

| Field | Library | Key Function Calls |
|-------|---------|--------------------|
| **Spectroscopy** | `specutils, astropy.modeling, astropy.io.fits, scipy.optimize, scipy.stats, lmfit, dust_extinction, astroML` | `Specutils.fitting.fit_lines, astropy.modeling.models.Voigt1D, astropy.io.fits.getval, scipy.optimize.curve_fit, scipy.stats.poisson.pmf,lmfit.Model, dust_extinction.shapes.P92, astroML.datasets.fetch_sdss_spectrum...` |
| **Photometry** | `photutils, lightkurve, astroquery.mast, astropy.table, astropy.stats, space_phot, regions` | `photutils.aperture.CircularAperture, lightkurve.search_lightcurve, lightkurve.RegressionCorrector, astroquery.mast.Catalogs.query_region, astroquery.mast.Tesscut.get_cutouts, astropy.stats.sigma_clipped_stats, space_phot.get_jwst_psf, regions.Regions.read...` |
| **Image Processing** | `numpy, scipy.ndimage, astropy.nddata, astropy.convolution, PIL, cv2, astrocut, wfc3tools, jwst` | `numpy.pad, numpy.histogram2d,scipy.ndimage.rotate, scipy.ndimage.shift,astropy.nddata.extract_array, astropy.convolution.Gaussian2DKernel, astropy.convolution.convolve,PIL.Image.open, cv2.resize, astrocut.fits_cut, wfc3tools.calwf3, jwst.datamodels.ImageModel...` |
| **Time Series Analysis** | `astropy.timeseries, lightkurve, pandas, scipy.signal, gatspy` | `astropy.timeseries.LombScargle, lightkurve.LightCurveCollection, lightkurve.TessTargetPixelFile, Pandas.to_datetime, scipy.signal.correlate, scipy.signal.correlation_lags, gatspy.periodic.LombScargleFast...` |
| **Cosmology & Large-Scale Structure** | `astropy.cosmology, healpy, astropy.coordinates, reproject, shapely` | `astropy.cosmology.LambdaCDM, healpy.mollview, healpy.ang2pix, astropy.coordinates.SkyCoord, astropy.coordinates.search_around_sky, reproject.reproject_interp, shapely.geometry.Polygon, shapely.geometry.Point...` |
| **Simulation & Modeling** | `astropy.modeling, keras, tensorflow, sklearn, webbpsf, acstools, stistools, refstis` | `astropy.modeling.fitting.LinearLSQFitter, keras.layers.Conv1D, keras.models.Model, tensorflow.GradientTape, tensorflow.reduce_mean, sklearn.ensemble.RandomForestRegressor, sklearn.decomposition.PCA, webbpsf.MIRI, acstools.focus_diverse_epsfs.psf_retriever, stistools.calstis.calstis, refstis.basedark.make_basedark...` |

Figure 5: Overview of astronomy-specific Python libraries and functions grouped by technical or topical field.

# A  Libraries and Function Calls in ASTROVISBENCH

Figure 5 shows the different categories of astronomy fields and their respective libraries that are included in ASTROVISBENCH. The benchmark covers a broad range of topics and commonly used packages to ensure that important tasks in the domain are represented.

# B Prompts Used for Benchmark Construction and Evaluation

**VLM Evaluation**  Prompt B.1 shows the instruction given to the judge model, as well as what information from each query execution is included. In addition to the instructions and example information, the VLM is given a JSON format output as an example of how to return judgments and rationales.

---

**B.1 Instructions to VLM for error judgments in automatic visualization evaluation**

**System:**  Your task is to evaluate the correctness and visual validity of the under-test data visualization related to astronomy that will be sent to you. You will return either "No Error", "Minor Error", or "Major Error" along with your rationale. The definitions of these errors are:

No Error: This indicates that this visualization conveys the same key information as the Ground Truth Visualization.

Minor Error: This indicates that this visualization could be fixed by making minor adjustments to the code or by clarifying under-specified details in the Visualization Query.

Major Error: This indicates that this visualization has a major deviation from the ground truth visualization, ultimately conveying very different information.

You will be given the visualization query that the visualization was created to fulfill, a "gold image" that is a completely correct fulfillment of the query that you can use to compare, and the corresponding "under-test" visualization created based on that requirement that you will assess the validity of.

These instructions must be followed when making your judgment: When you are evaluating a visualization, compare that visualization against the Ground Truth Visualization. You can also use the visualization query, and the code corresponding to the visualization query to inform your judgments. However, the main question you are being asked is: Does this visualization convey the same key information as the ground truth visualization?

In addition to the gold image and under-test image, you will receive the gold visualization code responsible for creating the visualization, and also the under-test code for the under-test visualization to help you analyze the differences. Note that this is a supplement and the bulk of your judgment should come from evaluating the images, because we are assessing what the images convey visually.

Please think carefully and provide your reasoning and score.
**Input:**
Visualization Query: {{Segment 1}}
Ground-Truth Code: {{Segment 2}}
Ground-Truth Visualization (Base64): {{Segment 3}}
Generated Code: {{Segment 4}}
Generated Visualization (Base64): {{Segment 5}}
**Output Format:**
"Rationale": "a brief reason", "Errors": "No Error", "Minor Error", or "Major Error",

---

**Notebook Distillation**  Prompt B.2 is used to convert a compressed chain of notebook cells containing a mixture of markdown and code cells into query-code pairs for the three sub-tasks (Setup, Processing, and Visualization) broken down from the task being described in the input notebook cells. The output from this prompt is then tested to ensure if the code present in it matches with the ground truth code from the notebook cells. If the output fails this test then new outputs will be regenerated until either that output is a match for the ground truth code or when the maximum number of regenerations is reached. Only query-code pairs in which this test passes are allowed to become a part of AstroVisBench.

---

**B.2 Notebook cell to code task distillation prompt**

**System:**
You are being provided with markdown and python code cells from a jupyter notebook. You need to convert this notebook into a special notebook assignment, without changing any of the code, using the following these guidelines:
- This assignment will be split into 3 sections
- The first section is the **setup** section. In this section, you will include only import statements and any

---

additional lines of code that sets up the enviornment or macros for this notebook
- The second section is the **processing** section. In this section, you will include any code that processes or analyzes data prior to visualization after setup.
- The third section is the **visualization** section. In this section, you will only include code that enables the end visualization as desired by the original notebook. The code in this section should output only ONE visualization.
- It is okay if there are visualizations made prior to the visualization section. This code should still be included.
- For each of these sections, you must include only ONE natural language query describing the code and ONE code cell which contains the code that corresponds to the query. The code must match to the code in the original notebook. Any deviation is UNACCEPTABLE.
- Only the code should be enclosed in tick marks ('''). There should only be 3 such code blocks, one for each section.
- Your output must end with the visualization code block.

Make sure to write your queries using these guidelines:
- Format your instructions like you are a astronomer needing help with the task. Talk like you are talking to a fellow astronomer who understands all the astronomy lingo. Don't simplify jargon or acronyms.
- Make sure your query is between 100 and 150 words.
- For the query describing the visualization, make sure the query naturally describes the provided output visualization in a way that is understandable and reproducible.
- Don't include specifics in your query (variable names, modules, packages, etc.). You are playing the role of someone who doesn't know much about Python.
- Don't be too vague as to leave too much room for interpretation. Your query should be such that the code in the corresponding *code cell* should be the only valid answer for it.
- Don't refer to cells or sections in your instructions. It should be formatted like a query to another person.

The queries you are generating should correspond to the code in the provided jupyter notebook. Indicate that you are doing this by:
- Filling in the code cells with the EXACT code from the provided jupyter notebook it corresponds to. Any kind of deviation is absolutely intolerable.
- If the original code cells are empty, do not bother writing anything down in these code cells.
- This can be checked by seeing whether all the code you have written combined is equivalent to all the code in provided notebook combined.

**Input:**
NOTEBOOK:

{{Compressed Task Notebook Cells}}

**Output Format:**

An assignment notebook with queries and code for the 3 stages inter-spliced. Code is specifically demarcated by being wrapped in tick marks (''').

**Underspecification Clarifications**    The prompt for generating clarifications for underspecifications within queries is shown in Prompt B.3. This prompt takes in as input the text query and the code associated with it and output a mapping of text spans in the query to strings and numerical values found in the ground truth code.

---

**B.3 Prompt for Clarifying Underspecifications**

**System:**
You will be given code and a text query which describes the task being performed by the code. Your task is to resolve underspecifications in this text query.
Instructions:
- Underspecifications are details which are necessary to reproduce the same result as the given code. These details are limited to string literals and numerical values found in the given code that are not specified either explicitly or implicitly in the text query.
- Format these underspecifications as follows. Separate each instance in a new line. In each line, highlight the span of text in the query that is underspecified followed by the value found in the code that clarifies it.

---

**Input:**
CODE:
{{code}}
TEXT QUERY:
{{query}}

**Output Format:**
Text span to value mappings that clarify underspecifications in the query.

**Code Generation**  Prompt B.4 is used to elicit code responses from LLMs to the queries present in ASTROVISBENCH. This main purpose of this prompt is to condition the model to only generate code in response to the query as opposed to a mix of code and natural language.

---

**B.4 Prompt for Querying Models to Generate Code**

**System:**
You are tasked with completing a jupyter notebook about astronomy. You will be given some markdown cells and some python code cells for context, and in response you must output only python code that accurately fulfills the goals of the notebook as described by the markdown text.

**Input:**
*For Processing Sub-task:*
{{Setup Query}}
{{Ground Truth Setup Code}}
{{Processing Query + Processing Underspecifications}}

*For Visualization Sub-task:*
{{Setup Query}}
{{Ground Truth Setup Code}}
{{Processing Query}}
{{Ground Truth Processing Code}}
{{Visualization Query}}

**Output Format:**
code generated for the respective sub-tasks

---

| Benchmark / Approach | Domain | Task | Focus |
|---|---|---|---|
| CodexEval [3] | General | Code Synthesis | General programming tasks |
| DS-1000 [20] | General | Library Use | Tool-grounded coding & data analysis |
| SWE-Bench [15] | SWE | Issue Resolution | Resolving real-world software issues |
| BigCodeBench [45] | General | Function Calling | Correctness of API calls |
| Gravity-Bench-v1 [19] | Physics | Scientific Discovery | Simulation-based scientific Q&A |
| PaperBench [33] | MLR | Workflow Reasoning | Automating steps in research papers |
| MLEBench [2] | MLR | Experiment Automation | ML experimentation pipeline tasks |
| SciCode [34] | Natural Sci. | Math Computation | Scientific mathematical computation challenges |
| EvaLLM [30] | General | Visualization Evaluation | JSON-level matching for plot evaluation |
| VisEval [4] | General | Visualization Evaluation | Readability and execution-based scoring |
| ChartCoder [42] | General | Chart-to-Code | Synthesizing datasets for chart generation |
| MatPlotAgent [40] | General | Agentic Code Gen. | Agentic, data-driven plot generation |
| DomainCQA [43] | Astronomy | Chart Understanding | Visual question answering on charts |
| CharXiv [37] | Natural Sci. | Chart Understanding | Reasoning over scientific chart data |
| AstroVisBench (ours) | **Astronomy** | **Research Workflows** | **End-to-end research assistance for astronomers** |

Table 6: Comparison of AstroVisBench with prior benchmarks and approaches. Our benchmark is unique in its focus on end-to-end, domain-specific scientific workflows in astronomy that require both specialized tools and visualization generation. Abbreviations: Software Engineering (SWE), ML Research (MLR).

## C   Related Work Table

Table 6 details a tabular summary of relevant related work compared to AstroVisBench. This benchmark distinguishes itself through its specialization in the astronomy domain as well as through its focus on evaluating LLMs on their ability to assist in scientific workflows, which require the use of specialized libraries and APIs and the need to visualize results according to domain-specific conventions.

Beyond just its value within the domain of astronomy, AstroVisBench uniquely targets an LLM's capability to implement research workflows to produce interpretable scientific visualizations, from which insights are derived. To elaborate:

**Existing generic coding benchmarks [1, 3, 16, 20, 24, 36, 39, 45]**   Compared to these benchmarks, AstroVisBench targets long-tail knowledge, focusing especially on the usage of domain-specific APIs and visualization generation.

**Scientific coding benchmarks [19, 33, 34, 2]**   Existing work has benchmarked models' ability to solve scientific computation problems [33], reproduce ML experiments as described from a small set of 20 papers [33], engage with a simulation using benchmark-specific tools [19], and solve problems in ML engineering competitions [2]. AstroVisBench differentiates itself from these benchmarks by evaluating whether models' can assist in a wide variety of tasks as a research assistant, aiding scientists amidst their own workflows when they do not know step-by-step workflows and may not know, in advance, the kinds of scientific utility a visualization would bring.

**Visualization benchmarks [4, 7, 12, 21, 26, 30]**   Many benchmarks exist in evaluating models' ability to generate visualizations. Most of these works focus on relatively simple visualizations (bar charts, line charts, etc.) with standard data formats, and they assess models' ability to follow highly explicit instructions. AstroVisBench, on the other hand, additionally evaluates whether models' are able to apply domain-specific knowledge to understand domain-adapted queries and interact with a variety of data formats to create diverse visualizations that comply with expert standards (see Figure 3).

## D   Evaluated Subject Models

We evaluated on a collection of open-source and closed-source LLMs, representing the latest models in each series at the time of this paper's writing.

- `GPT-4o` is an autoregressive "omni" model, accepting any combination of text, audio, image, and video and outputting any combination of text, audio, and image.

- `Claude 3.7 Sonnet` is a "hybrid reasoning model" that can accept inputs in different modalities.

- `Claude 4.0 Opus` is the latest and most capable of the Claude series of models trained for more advanced coding capabilities.

- `Qwen-2.5 (72B)` is the strongest open-source LLM at its size available at the time of this writing.

- `Llama-4 Maverick (17Bx128E)` is Meta's leading model, using an MoE architecture, and focuses on multimodality in text and image inputs.

- `Gemini 2.5 Pro` is Google's most advanced model to date, and obtains top results on most current benchmarks involving code generation, image understanding, and science.

- `o3-mini` is a smaller and more cost-efficient version of OpenAI's reasoning series models, with strong reported performance in science, math, and coding capabilities.

- `QwQ (32B)` is an open-source reasoning model specialized for math and coding.

**Hyperparameter Settings**   For the proprietary models that we evaluated (`GPT-4o`, `o3-mini`, `Claude 3.7 Sonnet`, `Claude 4.0 Opus`, `Gemini 2.5 Pro`), we used the default hyperparameters as defined by their respective APIs. For the open source models (`Qwen 2.5`, `Llama-4 Maverick`, `QwQ`), we used the Together.AI API with default hyperparameters. In terms of temperature, this means that all the models we evaluated were set with temperature of 1. The default `top_p` is also 1 for most models/APIs except for Gemini, where the default value is set at 0.95. There is no default value for the `max_tokens` hyperparameter in the Anthropic API, so we set it to be 1024 tokens for `Claude 3.7 Sonnet` generations. We checked the token counts of all the 864 responses generated by `Claude 3.7 Sonnet` and found that only ten responses ever hit this limit. For `Claude 4.0 Opus`, we additionally activated the extended thinking option to enable all its reasoning capabilities and set it to have `max_tokens` of 8000 while having 3000 tokens budgeted for thinking.

## E   Expert Judgment Rationales

Shown in Figures 6, 7, 8 are examples of astronomer annotations on correct, minor error and major error generations, respectively. Experts are given the original query, the ground truth visualization code and image, the generated visualization code and image, as well as the original notebooks. They then determine an error category as well as a justifying rationale. Visualizations that have "no error" can still have slight deviations from ground truth images, as shown in Figure 6. Mainly, the scientific utility of the visualization determines the error judgment. In Figure 7, it can be seen that while there are visual errors mentioned by the expert in the generated plot, the key scientific information being shown is still equivalent to the ground truth. Meanwhile, in Figure 8, there is an incorrect calculation being applied (as pointed out by the expert) that results in a plot that is very different from the ground truth. Because the scientific utility of the plot is compromised, this is a major error.

## F   Computation Resources

We ran the execution-based evaluation framework for AstroVisBench on a system using two 56-Core Intel Xeon MAX 9480 CPUs with 128GB of RAM in total. Running the execution-based framework for a single LLM also required around 100GB of storage, with the execution environment taking around 50GB of space while these remain 50GB is required for storing pickled Python objects resulting from the variable inspection test described in Section 3.1.

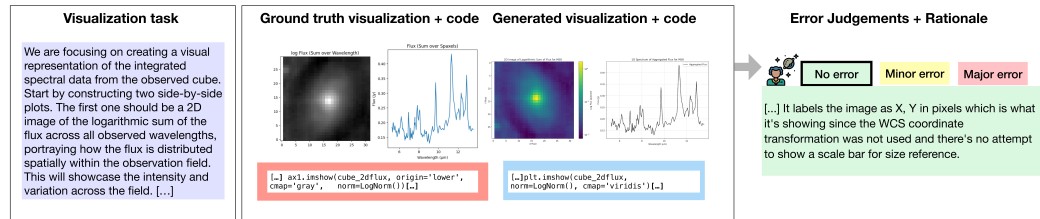

Figure 6: An example of expert judgment on a "no error" example. There are slight differences between the ground truth and generated visualizations, most notably an extra intensity gradient, but this does not detract from the scientific value of the visualization.

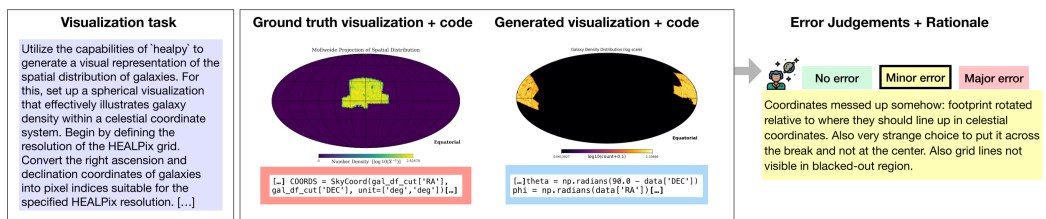

Figure 7: An example of expert judgment on a "minor error" example. The correct data are used but an incorrect transformation of coordinates has been applied causing the footprint to be shown off-center. The choice of color scheme makes it hard to see the grid lines, which was also considered as part of the judgment.

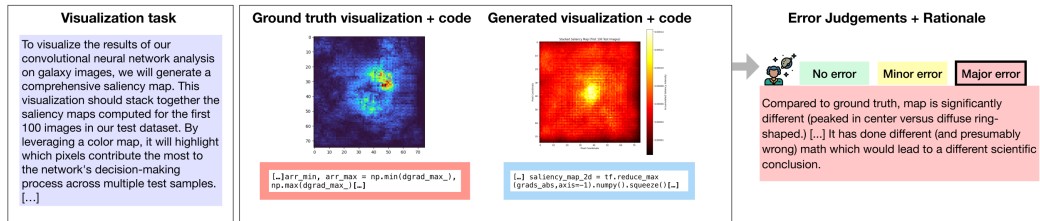

Figure 8: An example of expert judgment on a "major error" example. The code has performed a mathematical operation that is different from the ground truth, and as a result is not displaying the requested image.

## G   License Information

We gathered Jupyter notebooks from publicly-available collections curated by the Astro Data Lab and by the Space Telescope Science Institute (STScI). Most of these notebooks fall under the BSD-3 License with the exception of notebooks sourced from two repositories authored by STScI that do not have any license information attached.[4] However, STScI's content use policy asserts no claim to copyright for any material it produces as per its contract with NASA.[5] We release ASTROVISBENCH under the Creative Commons License-BY-3.

---

[4] https://github.com/spacetelescope/jdat_notebooks, https://github.com/spacetelescope/hellouniverse

[5] https://www.stsci.edu/copyright

