# OpenReview forum: "AstroVisBench: A Code Benchmark for Scientific Computing and Visualization in Astronomy"
_NeurIPS.cc/2025/Datasets_and_Benchmarks_Track — NeurIPS 2025 Datasets and Benchmarks Track poster_

### Official Review · Reviewer_EQ9S · 2025-06-29

**Rating:** 5
**Confidence:** 3

**Summary:**

This paper introduces the ASTROVISBENCH benchmark. It is the first benchmark designed to evaluate LLMs on scientific computing and visualization tasks in the domain of astronomy. The benchmark aims to evaluate the effectiveness of LLMs in generating domain-specific data analysis workflows and producing accurate and insightful visualizations.

**Additional Feedback:**

The benchmark is curated from existing Python notebooks. In this case, the actual novelty of the paper remains unclear, and it is still not evident how the proposed benchmark offers improved evaluation for LLMs.

**Dataset Code Accessibility:**

Yes

**Ethical Considerations:**

No, there are no or only very minor ethics concerns

**Final Justification:**

I keep my positive score for this paper. The authors should revise the paper to make all details clear.

**Limitations Weaknesses:**

1. There are also some methods, evaluations, or benchmarks discussed about scientific visualization tasks[1-2].  The authors should carefully discuss the novelty of the evaluation and proposed benchmarks.

2. Even the ASTROVISBENCH  benchmark is the first one for the domain of astronomy. However, I can not really tell apart the real difference between the challenges of the astronomy domain and other scientific domains. The authors claim that it is challenging for scientific computing and visualization. Can we show some error type classifications or cases to illustrate both points?

[1] ChartCoder: Advancing Multimodal Large Language Model for Chart-to-Code Generation.
[2] MatPlotAgent: Method and Evaluation for LLM-Based Agentic Scientific Data Visualization.

**Strengths Contributions:**

1. It is the first benchmark to evaluate LLMs on scientific computing and visualization tasks in the domain of astronomy.

2. The paper proposes an LLM based evaluation framework to validate against annotations from professional astronomers.

3. Experimental results reveal a significant performance gap in current LLMs' ability to assist with scientific discovery in astronomy.

4. ASTROVISBENCH offers a new way for developing LLM-driven workflows that are essential across many scientific fields.

---

> ### Author Rebuttal · Authors · 2025-07-31
>
> Thank you for reviewing our paper!
>
> > There are also some methods, evaluations, or benchmarks discussed about scientific visualization tasks[1-2]. The authors should carefully discuss the novelty of the evaluation and proposed benchmarks.
>
> Thank you for bringing these papers to our attention. We will make sure to cite them in the camera-ready version. AstroVisBench, like the approaches in these papers, does evaluate visualizations produced by code. What sets our benchmark apart is not just its domain specificity, but also its focus on evaluating the entire end-to-end process of generating domain-specific visualizations, which the methods in these papers, with their standardized data access and lack of domain focus, do not address. The model has to interact with different forms of data, interface with niche libraries to transform the data and visualize it as if it were an expert astronomer. AstroVisBench is not just evaluating domain-specific visualizations, but it is evaluating whether models are able to effectively assist in research workflows, in which being able to visualize experimental results is a core component.
>
> > However, I can not really tell apart the real difference between the challenges of the astronomy domain and other scientific domains. The authors claim that it is challenging for scientific computing and visualization. Can we show some error type classifications or cases to illustrate both points?
>
> In section 6, under the “Visualization Errors” paragraph, we describe the domain-specific errors made by LLMs while producing visualizations. These errors usually stem from a failure to follow in-domain conventions, such as mistaking the axes order for magnitude and sky coordinates, making inappropriate choices for axes scales, and failures to emphasize astronomical features in images while de-emphasizing noise. We also found that LLMs also struggle at times to interpret the type of visualization desired or necessary calculations needed to produce such a visualization. Similar issues can occur in any specialized domain, and AstroVisBench provides a model for benchmarking such issues given the subject matter expertise being available and involved in the construction of the benchmark.
> In general, there are real challenges unique to astronomy when it comes to visualizing data as opposed to other scientific domains. Because astronomy is fundamentally an observational science, it has an unique relationship with high-dimensional data and visualizations. The data themselves are often in the form of 2D  and 3D images, which lend themselves to graphical representation. These data are also multi-modal (e.g., multi-wavelength of different dimensions collected from different facilities), contain low-signal-to-noise features, and are large (e.g.,at peak the Vera Rubin Observatory will produce 20TB of data each night). Consequently, within astronomy visualizations are a critical means of efficiently exploring data relationships, classifying structures, and discovering new sources. Historically, due to the complexity of these data, insufficient flexibility of classical statistical methods, and lack of simple model descriptions, many analyses were (and are!) conducted ‘by eye.’ One such famous effort is the Galaxy Zoo project.  Today many of these workflows are being (imperfectly) automated by AI methods;  however, expert inspection remains the gold standard. Due to this data complexity, developments within the astronomy domain have important  implications for visualization challenges in a broad range of other scientific fields – fields where the data are generally simpler, smaller, and/or less accessible to AI models.

---

### Official Review · Reviewer_Zj35 · 2025-07-02

**Rating:** 5
**Confidence:** 4

**Summary:**

This paper introduces ASTROVISBENCH, a novel benchmark designed to evaluate large language models (LLMs) on end-to-end scientific computing and visualization tasks in the astronomy domain. It consists of 864 tasks extracted from 110 real-world Jupyter notebooks and tests LLMs on their ability to process astronomical data and generate scientifically valid visualizations. ASTROVISBENCH uniquely combines execution-based code validation and visualization evaluation using a validated vision-language model (VLM) aligned with expert astronomer judgments. The results highlight a significant gap in current LLM capabilities, particularly in using domain-specific libraries and producing accurate visual representations, offering critical insight into the development of LLMs as scientific assistants.

**Dataset Code Accessibility:**

Yes

**Ethical Considerations:**

No, there are no or only very minor ethics concerns

**Final Justification:**

After reading the authors' rebuttal, I believe the weaknesses I raised are not as serious as I initially thought. As the authors have adequately addressed my concerns, I will raise my rating to 5.

**Limitations Weaknesses:**

1. While I appreciate the authors provide AstroVisBench as a useful data resource, I'm a little concerned about the technical novelty of this work. The data construction pipeline (query generation, code construction, LLM-as-a-judge evaluation) seems to be straightforward. Also, the domain is limited to astronomy, and the task types are limited to visualization tasks (and the data processing part before visualizing them). There are actually previous benchmarks (such as [1]) that share similar ideas with this paper but include more domains and more task types. I think only focusing on astronomical visualization tasks might limit the scope and scalability of this work.

2. Following the previous point, the main data source to create AstroVisBench is some astronomical tutorials. I wonder how could we apply this paper's pipeline to other domains if we cannot find such a well-established tutorial collection? Is there a way to adopt more general and realistic data sources, such as the repositories of academic papers?

3. I'm a bit unclear about the domain specialty of this coding benchmark. Could the authors further explain a bit on this? Specifically, what's the uniqueness of astronomical visualization code compared with general visualization code? Is it just the difference of libraries being used (as listed in the paper), or what else?

**Strengths Contributions:**

1. This paper presents AstroVisBench, a useful resource to benchmark LLMs on end2end data processing and visualization tasks in astronomy domain. The paper clearly illustrated the steps of data construction. Also the LLM-as-a-judge part for evaluation is validated by domain-experts and show strong correlations.

2. The authors conduct experiments to show the performance of major state-of-the-art LLMs on astronomic visualization tasks, and provide an in-depth error analysis to demonstrate the bottlenecks of LLMs solving such tasks.

3. The paper is well-written and easy-to-follow.

---

> ### Author Rebuttal · Authors · 2025-07-31
>
> Thank you for reviewing our paper!
>
> > I'm a little concerned about the technical novelty of this work…There are actually previous benchmarks (such as [1]) that share similar ideas with this paper but include more domains and more task types. I think only focusing on astronomical visualization tasks might limit the scope and scalability of this work.
>
> Beyond just its value within the domain of astronomy, AstroVisBench uniquely targets an LLM’s capability to implement research workflows to produce interpretable scientific visualizations, from which insights are derived. To elaborate:
> Existing generic coding benchmarks [1, 2, 14, 19, 23, 35, 38, 42]: Compared to these benchmarks, AstroVisBench targets long-tail knowledge, focusing especially on the usage of domain-specific APIs and visualization generation.
> Scientific coding benchmarks [18, 32, 33, MLE-Bench (Shern et al. 2025)]: Existing work has benchmarked models’ ability to solve scientific computation problems [33], reproduce ML experiments as described from a small set of 20 papers [32], engage with a simulation using benchmark-specific tools [18], and solve problems in ML engineering competitions (Shern et al. 2025). AstroVisBench differentiates itself from these benchmarks by evaluating whether models’ can assist in a wide variety of tasks as a research assistant, aiding scientists amidst their own workflows when they do not know step-by-step workflows and may not know, in advance, the kinds of scientific utility a visualization would bring.
> Visualization benchmarks [3, 6, 11, 20, 25, 29]: Many benchmarks exist in evaluating models’ ability to generate visualizations. Most of these works focus on relatively simple visualizations (bar charts, line charts, etc.) with standard data formats, and they assess models’ ability to follow highly explicit instructions. AstroVisBench, on the other hand, additionally evaluates whether models’ are able to apply domain-specific knowledge to understand domain-adapted queries and interact with a variety of data formats to create diverse visualizations that comply with expert standards (see Figure 3).
>
> The benchmark is also formulated to span tasks in a range of astronomy subfields, including stellar and galactic astrophysics, cosmology, and transient detection (e.g., supernovae, stellar mergers, and asteroids). Tasks included in this benchmark have roughly corresponding tasks in other scientific disciplines, including visualizing structures in 3D MRI images, displaying particle signatures in large particle physics experiments, and identifying features/outliers in remote sensing.
>
> The complexity involved in visualizations for astronomy also pose an interesting challenge for models to replicate, as they can involve non-typical representations of spatial data. For example, consider the visualization depicted in Figure 7. These test cases serve to help develop better models more capable of presenting complex data with the appropriate level of spatial awareness. Please also refer to our response to the last point for a broader picture of the discipline, visualization and observations within astronomy, and broader impacts.
>
>
> > how could we apply this paper's pipeline to other domains if we cannot find such a well-established tutorial collection? Is there a way to adopt more general and realistic data sources, such as the repositories of academic papers?
>
> Thank you for asking this question! The methodology used in AstroVisBench can be easily applicable to any Jupyter notebook from any domain, as they are great examples of research workflows, often being used to explore data, perform experiments, and visualize results. To apply this to code repositories, similar workflows must be extracted from its contents, either through existing code in the repository or through synthetic code generated based on written descriptions of experiments. These workflows can then be adapted into tasks similar in style to the ones found in AstroVisBench. We did some preliminary analysis of this workflow and found that it would take some human effort to clean up the results, so we did not pursue it further for this work, but we think this is a promising direction for scaling the benchmark out.
>
> > What's the uniqueness of astronomical visualization code compared with general visualization code? Is it just the difference of libraries being used (as listed in the paper), or what else?
>
> There are some unique libraries used for astronomy visualization, such as healpy, reproject, and shapely. However, even when astronomers use more general libraries like matplotlib or seaborn, they use them to construct visualizations with domain-specific requirements. In our evaluation, we observed that LLMs oftentimes fail to meet these requirements in the visualizations they create. In section 6, in the visualization errors paragraph, we describe some of these domain-specific errors, such as mistaking the axes order for magnitude and sky coordinates, inappropriate choices for axes scales, and failures to emphasize astronomical features in images while de-emphasizing noise. In addition to these errors caused by a failure to follow in-domain convention, LLMs also struggle at times to interpret the type of visualization desired or necessary calculations needed to produce such a visualization.
>
> Astronomy, as a fundamentally observational science, has a unique relationship with high-dimensional data and visualizations. The data themselves are often in the form of 2D  and 3D images, which lend themselves to graphical representation. These data are also multi-modal (e.g., multi-wavelength of different dimensions collected from different facilities), contain low-signal-to-noise features, and are large (e.g.,at peak the Vera Rubin Observatory will produce 20TB of data each night). Consequently, within astronomy visualizations are a critical means of efficiently exploring data relationships, classifying structures, and discovering new sources. Historically, due to the complexity of these data, insufficient flexibility of classical statistical methods, and lack of simple model descriptions, many analyses were (and are!) conducted ‘by eye.’ One such famous effort is the Galaxy Zoo project.  Today many of these workflows are being (imperfectly) automated by AI methods;  however, expert inspection remains the gold standard. Due to this data complexity, developments within the astronomy domain have important  implications for visualization challenges in a broad range of other scientific fields – fields where the data are generally simpler, smaller, and/or less accessible to AI models.

---

### Official Review · Reviewer_kWsy · 2025-07-02

**Rating:** 4
**Confidence:** 3

**Summary:**

The paper introduces a first benchmark designed to evaluate the capabilities of LLMs in performing scientific computing and visualization tasks specific to the field of astronomy. The benchmark assesses an LLM's ability to generate Python code for two types of tasks: processing and analyzing astronomical data and creating domain-specific visualizations from that data. The tasks are sourced from real-world Jupyter notebooks used by professional scientists.

**Dataset Code Accessibility:**

Yes

**Ethical Considerations:**

No, there are no or only very minor ethics concerns

**Final Justification:**

I think the feedback has answered my questions well, and I give it a relatively positive rating.

**Limitations Weaknesses:**

LLMs are used to help generate the queries for the benchmark tasks. And this process may have introduced some noise into the benchmark. The evaluation of automated visualization relies heavily on a multimodal model as a judge. Although the judge is shown to correlate well with human experts on a subset of tasks, it could still produce judgments that are not perfectly aligned with expert opinion.

**Strengths Contributions:**

It is the first benchmark to specifically target both scientific computing and visualization within the astronomy domain. The benchmark's tasks are sourced from expert-level Jupyter notebooks, ensuring the tasks are representative of real-world scientific workflows.

I think that, compared to the problems with the dataset, the scenarios that the dataset is aimed at are more important. Initial attempts in the field of AI4Sci always need to first organize available benchmarking, and I think this is the value of this paper.

---

> ### Author Rebuttal · Authors · 2025-07-31
>
> Thank you for reviewing our paper!
>
> > LLMs are used to help generate the queries for the benchmark tasks. And this process may have introduced some noise into the benchmark.
>
> Processing the original notebooks automatically with a strong LLM is a tradeoff we made for the scalability and generalizability of our method. We found that GPT-4o is effective in grounded generation, and we did have expert astronomers verify a random subset of 135 queries out of the 864 total queries, in which they did not find issues that involved hallucinations that would alter the corresponding implementation of the original notebook instructions.
>
> More specifically, we see the following as the most salient possible failure modes:
> 1. The model hallucinated new content for the query. Very unlikely due to the function of these models and the prompt they were given.
> 2. The model didn't provide a precise enough query. We did analysis to make sure this didn't happen much, although it still happened; see Section 2 “Query Generation Disiderata”, specifically about addressing underspecifications.
> 3. The query doesn't capture the original meaning. We verified with our experts that this does not happen in the subset of 135 that they verified.
> We acknowledge that this still remains a limitation of our method and we have addressed it as such in the limitation section of our paper (Lines 358-360).
>
> > The evaluation of automated visualization relies heavily on a multimodal model as a judge. Although the judge is shown to correlate well with human experts on a subset of tasks, it could still produce judgments that are not perfectly aligned with expert opinion.
>
> While we acknowledge that LLM-as-a-judge is not always perfect, always having an expert to judge every solution is infeasible for a benchmark where new solutions can be continuously submitted. Thus, one has to resort to some sort of automatic evaluation as in almost all of the existing benchmarks. The evaluation scheme, though on the surface simple, is a result of 6 hours of discussion across 5 senior expert astronomers, plus additional individual annotation time. These insights also helped us craft the VLM judge prompt. The VLM judgment correlation with human experts’ average/majority labels reached 0.82 Spearman (Table 2), which is actually higher than the pairwise expert-expert correlation (0.69, Table 1).

---

### Official Review · Reviewer_sCK2 · 2025-07-03

**Rating:** 5
**Confidence:** 4

**Summary:**

- The authors introduce AstroVisBench, a novel benchmark for evaluating LLMs on end-to-end astronomy workflows, from data processing to domain-specific visualizations.
- The benchmark comprises 864 tasks (432 processing, 432 visualization) adapted from 110 Jupyter notebooks, uses execution-based checks for processing tasks, and a vision-LLM judged against expert annotations for visualization quality.
- Seven state-of-the-art models are evaluated, revealing high failure and major error rates, indicating significant gaps in current LLMs’ scientific capabilities.

**Dataset Code Accessibility:**

Yes

**Dataset Code Comments:**

Authors have provided links to the code and documentation.

**Ethical Considerations:**

No, there are no or only very minor ethics concerns

**Final Justification:**

Since all of my concerns are addressed with additional ablation studies, I am raising my score.

**Limitations Weaknesses:**

- In the related work section, incorporating a comparison table that highlights how this benchmark’s features differ from existing scientific datasets would further strengthen the paper’s positioning. While the domain focus is certainly valuable, it would be helpful to explicitly showcase what additional capabilities, beyond domain specificity, this benchmark brings to evaluate recent vLLMs.
- Relying on GPT-4o to both segment notebooks into distinct stages and generate corresponding natural-language queries introduces potential biases and hallucinations. The paper would benefit from a more detailed evaluation of GPT-4o’s query-generation accuracy and its challenges. Additionally, discussing how the choice of a different LLM might impact data quality and downstream evaluation of models and findings.
- I may have missed it, but it seems key evaluations are conducted in the zero-shot setting for LLMs. I think the evaluation of recent advanced prompting techniques, such as Image-of-Thought, should be conducted. Because it is very obvious that current models cannot perform well in a specialized domain, as they lack particular knowledge. Maybe some experiments with a few-shot setting can also give more insights into model performance. A small study on prompt sensitivity could be helpful.
- While expert judgments validate the VLM-judge, preference agreement is low (0.44). It’s unclear how this subjectivity propagates to the automatic scoring on the full benchmark.
- Please specify the temperature and all other hyperparameter settings for the proprietary models to ensure reproducibility.

**Strengths Contributions:**

- Evaluating vision-LLMs on scientific tasks is both timely and important, and this paper makes a significant contribution in that direction.
- This work explores a vision-LLM judge for evaluation, which is valuable and in line with the latest research trends about using LLMs effectively for verification and evaluation.
- Detailed error analysis provides a better understanding of the strengths and limitations of recent vLLMs, which, in my opinion, is valuable for future development of vLLMs.

---

> ### Author Rebuttal · Authors · 2025-07-31
>
> Thank you for your review and feedback!
>
> **Re related work:**
> Thank you for this suggestion. We will happily include a table. Beyond just its value within the domain of astronomy, AstroVisBench uniquely targets an LLM’s capability to implement research workflows to produce interpretable scientific visualizations, from which insights are derived. To elaborate:
> Existing generic coding benchmarks [1, 2, 14, 19, 23, 35, 38, 42]: Compared to these benchmarks, AstroVisBench targets long-tail knowledge, focusing especially on the usage of domain-specific APIs and visualization generation.
> Scientific coding benchmarks [18, 32, 33, MLE-Bench (Shern et al. 2025)]: Existing work has benchmarked models’ ability to solve scientific computation problems [33], reproduce ML experiments as described from a small set of 20 papers [32], engage with a simulation using benchmark-specific tools [18], and solve problems in ML engineering competitions (Shern et al. 2025). AstroVisBench differentiates itself from these benchmarks by evaluating whether models’ can assist in a wide variety of tasks as a research assistant, aiding scientists amidst their own workflows when they do not know step-by-step workflows and may not know, in advance, the kinds of scientific utility a visualization would bring.
> Visualization benchmarks [3, 6, 11, 20, 25, 29]: Many benchmarks exist in evaluating models’ ability to generate visualizations. Most of these works focus on relatively simple visualizations (bar charts, line charts, etc.) with standard data formats, and they assess models’ ability to follow highly explicit instructions. AstroVisBench, on the other hand, additionally evaluates whether models are able to apply domain-specific knowledge to understand queries that are often not explicit (since users may not know detailed steps and often presuppose in-domain common knowledge) and interact with a variety of data formats to create diverse visualizations that comply with expert standards (see Figure 3).
>
> **Re GPT-4o to make queries:**
> Processing the original notebooks automatically with a strong LLM is a tradeoff we made for the scalability and generalizability of our method. We found that GPT-4o is effective in grounded generation, and we did have expert astronomers verify a random subset of 135 queries out of the 864 total queries, in which they did not find issues that involve hallucinations that would alter the corresponding implementation of the original notebook instructions.
>
> More specifically, we see the following as the most salient possible failure modes:
> 1. The model hallucinated new content for the query. Very unlikely due to the capabilities of these models and the prompt they were given, which constrains them primarily to rewrite.
> 2. The model didn't provide a precise enough query. We did analysis to make sure this didn't happen much, although it still happened; see Section 2 “Query Generation Disiderata”, specifically about addressing underspecifications.
> 3. The query doesn't capture the original meaning. We verified with our experts that this does not happen in the subset of 135 that they verified.
> We acknowledge that this still remains a limitation of our method and we have addressed it as such in the limitation section of our paper (Lines 358-360).
>
> **Re different LLM to make queries:**
> We did preliminary examinations of other SOTA LLMs’ ability to generate instructions, and we found that GPT-4o was not only the more cost effective option, but it also followed instructions better and generated more realistic queries. Gemini 2.5 Pro was a close contender, but it was overly descriptive, leaking domain knowledge the model was expected to know, and it was more expensive as well.
>
> **Re Prompting Techniques:**
> We have added some results on few-shot prompting below. There are three reasons why we focused on zero-shot prompting in our evaluation: (1) The context provided for a task is often very long, making few-shot prompting impractical. (2) As models advance, we see implementation moving away from few-shot prompts. For instance, agentic harnesses often use zero-shot prompts (e.g., MatPlotAgent/MatPlotBench, CodeBenchGen, PaperBench, SciCode). (3) Lastly, the long context we provide does give the model relevant knowledge pertaining to the task. This context also includes natural language and ground truth code from prior stages in the task (see section 2.1). We gave this information to ensure that the generated code, when executed, could be comparable to the ground truth.
>
> It is possible to use techniques such as self-refinement to correct errors, or modified prompts to drive more advanced reasoning (e.g., budget-forcing as in s1). However, we ultimately decided to evaluate more basic setups as these are often reflective of how scientists use these models to begin with (anecdotally from our discussions with astronomers). Similarly, Image-of-Thought doesn’t readily apply here because the input does not contain images from the get-go, though it could be used as part of an iterative refinement process.
>
> **Re few-shot experiments:**
> As per your request, we did experiment with adding two few-shot examples with the claude-sonnet-4 model, as it was one of the few models with a large enough context window. We report the results below:
>
> | Model | PCrash % | VIscore | VCrash % | VisFail % | NoE % | MiE % | MaE % |
> |---|---|---|---|---|---|---|---|
> | claude-sonnet-3.7 | 50.9 | 0.633 | 34.7 | 21.3 | 9.5 | 14.6 | 19.2 |
> | few-shot-claude-sonnet-4 | 51.9 | 0.580 | 33.1 | 7.2 | 9.7 | 30.1 | 19.9 |
>
> Here we show the percent of test instances that crashed (PCrash %/ VCrash%) for each type of task, the variable inspection score (VIscore), the percent of instances where the model failed to produce only one visualization (VisFail %), and the breakdown of no error (NoE %), minor error (MiE %) and major errors (MaE %), the same metrics reported for the automatic visualization evaluation (see Table 5 and caption). Adding few-shot examples had little effect for the most part, though VisFails decreased significantly while the number of visualizations with minor errors increased in tandem.
>
> **Re prompt sensitivity:**
> As per your request, we also did a small-scale prompt sensitivity analysis. We focused on five visualization task queries from the benchmark and qualitatively analyzed the resulting visualization produced by claude-sonnet-4 after modifying the prompt in the following ways: (1) attributing a persona to the model (“you are an expert astronomer…”), (2) adding compilation and aesthetic requirements (“The code needs to compile and the visualizations need to be aesthetically pleasing”), (3) fully paraphrasing the query, and (4) rewriting the query in an entirely different style (query re-rewritten in the style of southern cowboy). For the last two variants we used ChatGPT to do the paraphrasing and re-writing.
>
>
>
> We evaluated the responses to these queries using our automatic evaluation framework and we present our results below, where we show the percent breakdown for each category.
>
> | Query | VCrash % | VisFail % | NoE % | MiE % | MaE % |
> |---|---|---|---|---|---|
> | Original | 20 | 0 | 40 | 40 | 0 |
> | Persona (1) | 20 | 40 | 0 | 20 | 20 |
> | Requirements (2) | 20 | 0 | 40 | 20 | 20 |
> | Paraphrase (3) | 0 | 20 | 20 | 0 | 40 |
> | Style Change (4) | 0 | 60 | 0 | 0 | 40 |
>
>
> Modifications 1-2 were designed to help the model, but these modifications did not make any improvements to the quality of visualizations as compared to the original query: Adding an “expert persona” led to more crashes and adding requirements led to one more instance of major error. Paraphrase prompts and stylistic changes (3-4) led to fewer crashes but increased the % of VisFail and major errors; this makes sense given that vocabulary changes (“comparative” vs “side-by-side”) or stylistic changes modify what is expected of the model from the original query. Ultimately, a larger scale study should be conducted to analyze the impact of different changes to queries across the benchmark. However, such a study is beyond the scope of this paper and so we leave it as a topic of future work.
>
> **Re preference agreement:**
> The preference agreement (which was not used in the final benchmark scoring scheme) is moderately low, but it is important to note that these preference judgments are only made when the expert annotator gives both LLM-generated visualizations the same error rating. We used these judgments to analyze if there was a consistent pattern beyond the error assessment that would make experts prefer one visualization over another. However, we found that any kind of preference was mainly subjective, depending on the expert’s background and aesthetic preferences. The automatic scoring, on the other hand, is based on the error rating described in section 3.2, and when we measure the correlation of the automatic scores against human judgments, we only compare against the error judgments.
>
> **Re hyperparameter settings:**
> For the proprietary models that we evaluated (GPT-4o, o3-mini, Claude Sonnet 3.7, Gemini 2.5 Pro), we used the default hyperparameters as defined by their respective APIs. For the open source models (Qwen 2.5, Llama 4 Maverick, QWQ), we used the Together.AI API with default hyperparameters. In terms of temperature, this means that all the models we evaluated were set with temperature of 1. The default top_p is also 1 for most models/APIs except for Gemini, where the default value is set at 0.95. There is no default value for the max_tokens hyperparameter in the Anthropic API, so we set it to be 1024 tokens for all generations. We checked the token counts of all the 864 responses generated by Claude Sonnet 3.7 and found that only ten responses ever hit this limit. We will make sure all of this information is included in our final version.

---

> > ### Comment · Reviewer_sCK2 · 2025-08-07
> >
> > Thank you for the detailed rebuttal as well as additional experiments. Please add discussions around them in the revised version. I revised my score accordingly.

---

### Decision · Program_Chairs · 2025-09-18

**Decision:**

Accept (poster)

**Comment:**

The authors introduce AstroVisBench, a novel benchmark for evaluating large language models (LLMs) on end-to-end astronomy workflows, spanning from data processing to domain-specific visualizations. The benchmark consists of 864 tasks (432 processing, 432 visualization) adapted from 110 Jupyter notebooks. It employs execution-based checks for processing tasks and a vision-LLM evaluated against expert annotations for visualization quality. Seven state-of-the-art models were evaluated, revealing high failure rates and significant error margins, highlighting substantial gaps in the scientific capabilities of current LLMs.

Reviewers initially raised concerns regarding the novelty of the benchmark, the differences between the challenges in the astronomy domain versus other scientific fields, and other related issues. However, these concerns were adequately addressed in the rebuttal. Therefore, I recommend accepting the paper.